# Unique Gut Microbiome Signatures among Adult Patients with Moderate to Severe Atopic Dermatitis in Southern Chinese

**DOI:** 10.3390/ijms241612856

**Published:** 2023-08-16

**Authors:** Yiwei Wang, Jinpao Hou, Joseph Chi-Ching Tsui, Lin Wang, Junwei Zhou, Un Kei Chan, Claudia Jun Yi Lo, Pui Ling Kella Siu, Steven King Fan Loo, Stephen Kwok Wing Tsui

**Affiliations:** 1School of Biomedical Sciences, Faculty of Medicine, The Chinese University of Hong Kong, Hong Kong; evywang2017@gmail.com (Y.W.); linwanggirl@gmail.com (L.W.); 2Centre for Microbial Genomics and Proteomics, The Chinese University of Hong Kong, Hong Kong; 3Microbiome Research Centre, BioMed Laboratory Company Limited, Hong Kong; cctsui_2013@hotmail.com (J.C.-C.T.); waynezhou@biomed.com.hk (J.Z.); yukichanuk10@gmail.com (U.K.C.);; 4Hong Kong Institute of Integrative Medicine, Faculty of Medicine, The Chinese University of Hong Kong, Hong Kong; 5Dermatology Centre, CUHK Medical Centre, The Chinese University of Hong Kong, Hong Kong; 6Hong Kong Bioinformatics Centre, The Chinese University of Hong Kong, Hong Kong

**Keywords:** gut microbiota, atopic dermatitis, 16S rRNA amplicon, gut–skin axis

## Abstract

Imbalance of the immune system caused by alterations of the gut microbiome is considered to be a critical factor in the pathogenesis of infant eczema, but the exact role of the gut microbiome in adult atopic dermatitis (AD) patients remains to be clarified. To investigate the differences of the gut microbiome between adult AD patients and healthy individuals, stool samples of 234 adults, containing 104 AD patients and 130 healthy subjects, were collected for 16S rRNA gene amplicon. Altered structure and metabolic dysfunctions of the gut microbiome were identified in adult AD patients. Our results illustrated that the adult AD patients were more likely to have allergies, particularly non-food allergies. In addition, the gut microbiome composition of the AD and normal groups were considerably different. Moreover, *Romboutsia* and *Clostridi-um_sensu_stricto_1* was enriched in the normal group, whereas *Blautia*, *Butyricicoccus*, *Lachnoclostridium*, *Eubacterium_hallii_group*, *Erysi-pelatoclostridium*, *Megasphaera*, *Oscillibacter*, and *Flavonifractor* dominated in the AD group. Additionally, purine nucleotide degradation pathways were significantly enriched in the AD group, and the enrichment of proteinogenic amino acid biosynthesis pathways was found in the normal group. This study provides insights into new therapeutic strategies targeting the gut microbiome for AD and evidence for the involvement of the gut–skin axis in AD patients.

## 1. Introduction

Atopic dermatitis (AD), also commonly referred to as eczema, is one of the most common skin diseases, characterized by recurrent chronic eczematous lesions, occurring with dry skin, pruritus, and obvious itching [1]. In the past decade, the global incidence of AD has been increasing, and the affected population involves all age groups ranging from infants to the elderly. This seriously influences the patient’s quality of life and brings a huge burden to healthcare resources [1]. AD patients usually may also be accompanied by other atopic diseases, such as allergic asthma and allergic rhinoconjunctivitis [2]. The pathogenesis of AD is not yet clear. It is generally believed that AD may be related to genetics, environmental factors, immune abnormalities, and abnormal skin barrier function. It is the interaction between genetic factors and environmental factors that mediates the occurrence and development of AD through immune pathways [3].

The gut microbiome is renowned as the second human genome, the homeostasis of which has been evidenced to have a beneficial effect on maintaining human health and is commonly affected by many factors, such as age, gender, diet, mood, and health status [4,5]. The gut microbiota of adults is proved to be more complex than that of infants [6]. Imbalance of the gut microbiome in early childhood precedes the onset of AD [3]. Numerous studies have focused on the characterization of the gut microbiome in AD infants or children because of the strong impact of the gut microbiome on the development and maintenance of the immune system in early life [7,8,9,10,11,12]. In the intestines of AD infants, the relative abundance of *Bifidobacterium*, *Enterococcus*, *Clostridium*, *Lactobacillus paracasei*, and *Ruminococcaceae* decrease [8,12,13]. In contrast, gut colonization with *Staphylococcus*, *Clostridia*, and *Feacalibacterium prausnitzii* is more prevalent in AD infants [14,15]. However, there are only a few studies that have clarified the role of the gut microbiome in adult AD patients.

In this study, we performed amplicon sequencing and revealed the differences in gut microbiome composition, biodiversity, and functional profiling using amplicon sequencing between healthy adults and AD patients from a large Hong Kong cohort.

## 2. Results

### 2.1. Characteristics of Participants

A total of 104 AD patients and 130 healthy subjects were recruited in this study, 143 of which provided the demographic data themselves by filling out the questionnaires. Parameters including age, sex, allergy, diarrhea, constipation, weight, and height were provided by the participants themselves. The original data can be obtained through the link https://github.com/evy-yiweiwang999/AD_gutmicrobiome_case_control/blob/main/metainfo.csv (accessed on 23 May 2023). Baseline characteristics of participants are summarized in Table 1 and Appendix A. A higher proportion of participants with a medical history of allergy (*p* = 0.0002), non-food allergy (*p* = 0.0301) especially, was observed among the AD group.

### 2.2. Distinct Gut Microbiome Composition between AD Patients and Healthy Subjects

To dissect potential dysbiosis of gut bacterial communities among AD patients, we compared the alpha and beta diversity between the AD group and normal group. Although, no significant difference (Shannon diversity index, *p* = 0.285) in alpha diversity, including richness and evenness of the gut microbiome, was obtained between the AD group and the normal group (Figure 1A,B, Appendix A). Beta diversity analysis results demonstrated that gut microbiome of the AD group was immensely distinct from that of the normal group based on Bray–Curtis (*p* = 0.001), Jaccard (*p* = 0.001), and unweighted-UniFrac (*p* = 0.025) distance metrics (Figure 1C, Appendix A). Furthermore, statistically significant differences in beta diversity based on Bray–Curtis (*p* = 0.001), Jaccard (*p* = 0.001), and unweighted-UniFrac (*p* = 0.031) were obtained between normal and severe AD groups, as well as normal and mild AD groups (Figure 1D,E, Appendix A). Collectively, these findings indicate differences in the gut microbial community structure between patients with AD and healthy individuals.

### 2.3. Gut Microbial Composition in Healthy Individuals and AD Patients

At the phylum level, a total of 15 phyla, including 12 from the kingdom of bacteria and 3 from archaea, were detected, which constituted the gut microbiome of both patients with AD and healthy individuals (Figure 2A, Appendix A), and the top 6 most abundant phyla accounted for over 99% of sequences in the dataset (Figure 2B). In both the AD and normal groups, *Firmicutes* (AD, 57.4%; normal, 58.2%) was dominant, followed by *Bacteroidota* (AD, 32.4%; normal, 32.1%), *Actinobacteriota* (AD, 6.2%; normal, 5.5%), *Proteobacteria* (AD, 2.7%; normal, 2.4%), *Desulfobacterota* (AD, 0.5%; normal, 0.7%), and *Fusobacteriota* (AD, 0.3%; normal, 0.5%; Figure 2B, Appendix A). In addition, by conducting ANCOM analysis, the gut microbial community of the AD group demonstrated a similar composition to the normal group at the phylum level (Appendix A). Additionally, no significant difference (AD vs. normal, *p* = 0.7566; Mild_AD vs. Severe_AD vs. normal, *p* = 0.5748) in *Firmicutes* to *Bacteroidetes* (F/B) ratio was detected across groups (Appendix A). At the genus level, the topmost abundant were *Bacteroides*, *Prevotella*, *Blautia*, *Faecalibacterium*, and *Bifidobacterium* (Figure 2C) in both groups. Furthermore, only the relative abundance of the *Clostridium_sensu_stricto_1* genus in the normal group was significantly higher than that in the AD group (W = 218) using ANCOM (Figure 2D).

### 2.4. Identification of Gut Microbial Signatures to Differentiate between AD and Healthy Subjects and Related to the Severity of AD

A total of 13 genera were identified as gut bacterial signatures at the genus level between AD patients and healthy individuals using the method of LEfSe (Figure 3, Appendix A). Nine genera, including *Blautia*, *Butyricicoccus*, *Lachncoclostridium*, *Eubacterium_hallii_group*, *Erysipelatoclostridium*, *Megasphaera*, *Oscillibacter*, *Flavonifractor*, and unclassified genera from the bacteria family of *Oscillospiraceae* were significantly enriched in the AD group. The other four genera, containing *Romboutsia*, *Clostridium_sensu_stricto_1,* and unclassified genera from two bacteria families, *Butyricicoccaceae* and *Erysipelotrichaceae,* were considerably enriched in the normal group (Figure 3A and Appendix A). Additionally, the relative abundances of *Blautia* and *Butyricoccus* were significantly positively correlated with the severity of AD. As the severity of AD deepens, the relative abundance of two genera, *Romboutsia* as well as *Clostridium_sensu_strico_1*, and one family, *Erysipelotrichaceae,* decreases significantly (Figure 3B). Additionally, the enrichment of *Bacteroides*, *Butyricicoccus*, and *Lachnospiraceae_UCG_008* was notably obtained in the Mild_AD group, while the relative abundance of *Blautia* in the Severe_AD group was significantly higher than that in the normal and Mild_AD groups (Appendix A).

In order to find the gut microbial signatures related to AD severity, Spearman correlation analysis was performed between the EASI score and different phylogenetics level of gut microbial compositions. *Fusobacteriota* at the phylum level, *Coriobacteriia* and *Fusobactriia* at the class level, *Alphaproteobacteria*.*_o*, *Coriobacteriales*, and *Fusobacteriales* at the order level, and *Veillonellaceae*, *Anaerofustaceae*, *Alphaproteobacteria._o._f*, *Butyricicoccaceae*, *Coriobacteriaceae*, *Fusobacteriaceae*, *Burkholderiaceae*, *Coriobacteriales_Incertae_Sedis*, and *Neisseriaceae* at the family level significantly correlated with EASI score. Moreover, 33 genera were identified to be remarkably associated with EASI score (Figure 4). In total, 4 of 33 genera, including *f_Erysipelotrichaceae;g_UCG_003*, *Romboutsia*, *Butyricicoccus*, and *Blautia,* were also selected as the gut microbial signature using the LEfSe analysis (Figure 3B).

### 2.5. Significantly Altered MetaCyc Pathways of the Gut Microbiome Associated with AD

Functional prediction analysis was performed on the gut metagenome using PICRUSt2 and LEfSe algorithms. This identified the difference in the function of the gut microbiome between AD patients and healthy subjects. Distinct MetaCyc pathways differences between AD and normal groups were calculated by LEfSe analysis on PICRUSt2 output (Figure 5, Appendix A). Specifically, dTDP-N-acetylthomosamine biosynthesis (PWY-7315), D-fructuronate degradation (PWY-7242), purine nucleobases degradation I (anaerobic) (P164-PWY), formaldehyde assimilation II (RuMP Cycle) (PWY-1861), superpathway of glycolysis and Entner–Doudoroff (GLYCOLYSIS-E-D), superpathway of N-acetylneuraminate degradation (P441-PWY), methanogenesis from acetate (METH-ACETATE-PWY), guanosine nucleotides degradation III (PWY-6608), purine nucleotides degradation II (aerobic) (PWY-6353), adenosine nucleotides degradation II (SALVADEHYPOX-PWY), and palmitate biosynthesis II (bacteria and plants) (PWY-5971) were enriched in AD patients. In contrast, urea cycle (PWY-4984), nitrate reduction VI (assimilatory) (PWY490-3), GDP-mannose biosynthesis (PWY-5659), glycolysis I (from glucose 6-phosphate) (GLYCOLYSIS), L-lysine biosynthesis II (PWY-2941), superpathway of L-tyrosine biosynthesis (PWY-6630), as well as superpathway of L-phenylalanine biosynthesis (PWY-6628) were identified to be enriched in the normal group (Figure 5A, Appendix A). Interestingly, most functional pathways (8/11) enriched in the AD group were assigned to the degradation/utilization/assimilation superclass, of which four pathways were affiliated with the purine nucleotide degradation class. Among the pathways enriched in the normal group, more than half (4/7) of the pathways were subjected to the superclass of biosynthesis, and three pathways belong to the class of proteinogenic amino acid biosynthesis (Figure 5A). In addition, we also compared the differences in the functional potential of the gut microbiome between Mild_AD and Severe_AD patients. Purine nucleotides degradation II (aerobic) (PWY-6353) and adenosine nucleotides degradation II (SALVADEHYPOX-PWY), belonging to purine nucleotides degradation, were found to be enriched in the Severe_AD group. By contrast, pyridoxal 5′-phosphate biosynthesis I (PRYIDOXSYN-PWY) and superpathway of pyridoxal 5′-phosphate biosynthesis and salvage (PWY0-845) pertaining to vitamin B6 biosynthesis were most abundant in the Mild_AD group (Figure 5B, Appendix A).

## 3. Discussion

To the best of our knowledge, this study is currently the first large-scale community-based gut microbiome study on adult atopic dermatitis. We evaluated the association between the incidence of adult AD and age, gender, BMI, the prevalence of allergies, and gastrointestinal symptoms. We also performed amplicon sequencing to investigate the characteristics of adult AD patients’ gut microbiomes from several aspects including gut microbial composition, biodiversity, and functional potential.

Our results support previous studies indicating that the incidence of allergy among AD patients is higher than that of healthy individuals. One previous study reported that nearly 80% of AD children will develop asthma or allergic rhinitis [16]. Several lines of evidence suggested that AD patients are more sensitive to aeroallergens and food allergens, which play an important pathogenetic role in the development of AD [2,17,18]. In our study, AD patients have higher rates of non-food allergies compared with the healthy subjects, which is consistent with the previous conclusion that adults and children over 5 years old appeared to be more sensitive to non-food allergens such as house dust mite [19,20]. However, we failed to obtain a difference in the prevalence of food allergy between AD and normal groups. Actually, associations between AD and food allergy remain controversial. Eller E et al. and Hon KL et al. reported that infants and toddlers with AD are more prone to food allergies [17,21]. Moreover, among AD patients, the rate of food allergy occurrence is ranging from 30% to 80%, depending on the population, however, the actual incidence of confirmed food allergies is much lower [17,22,23]. Furthermore, a previous study proved that immune response to some of the particular microbial PG might play a vital role in allergy type response in the patient [24]. Although Carol C et al. reported that children with atopic eczema had a higher incidence of diarrhea, especially in those with diffuse eczema, we did not observe differences in the frequency of gastrointestinal symptoms between adult AD patients and healthy individuals. Consistent with our results, their study also failed to obtain an increase in constipation in AD children [25]. These inconsistent results may be attributable to differences in age, geographical locations, and scales of assessing gastrointestinal symptoms.

Previous research conclusions on alpha diversity of the gut microbiome in AD patients were controversial. In our study, we found that the gut microbial community of adult AD patients displayed a similar alpha diversity. Similarly, five studies identified no significant differences in the alpha diversity of the gut microbiome in AD infants or children compared with healthy control counterparts [8,10,11,13,26]. Lee and colleagues reported a higher alpha diversity in infant atopic eczema [11]. In contrast, three other studies indicated that the alpha diversity of infants and young toddlers with eczema or at high risk for AD decreased compared to controls [7,9,27]. In terms of beta diversity, although after trying different data preprocessing methods using mbDenoise [28] and mbImpute [29], it is still impossible to distinguish the two groups in the scatter plot. However, no matter what data preprocessing methods we used, we can still obtain a statistically significant difference in beta diversity.

We confirmed the difference in the composition of the intestinal bacteria of AD and normal groups and pointed out the potential microbial signatures. Our results demonstrated that compared to the normal group, the relative abundance of two bacterial genera, *Clostridium_sensu_stricto_1* and *Romboutsia*, were reduced in AD patients. *Clostridium_sensu_stricto_1* was the only species with a differential abundance test that was consistently obtained by the two algorithms of ANCOM and LEfSe. *Clostridium_sensu_stricto_1,* containing 160 species [30], includes *Clostridium butyricum*, which has a strong ability to produce butyric acid, a type of short-chain fatty acid (SCFA) [31]. Similarly, *Romboutsia sedimentorum* in the genus *Romboutsia* is evidenced to produce short-chain fatty acids such as acetic acid, isobutyric acid, and isovaleric acid by fermenting glucose [32]. Short-chain fatty acids, which play an important role in the gut–skin axis, can stamp out the immune responses by inhibiting the cytokine production, migration, proliferation, and adhesion of inflammatory cells. SCFAs can also regulate the apoptosis and activation of immune cells by deactivation of NF-κB signaling pathways and inhibiting histone deacetylase, which promotes the cell proliferation that regulates various skin physiological functions, for example, regulating the differentiation of hair follicle stem cells and wound healing [33]. Moreover, SCFAs can provide energy for the intestinal epithelium and improve the integrity of the intestinal epithelial barrier to prevent microorganisms and toxins from entering the body fluid circulation to trigger Th2 immunity, and further resulting in a disturbance of the skin homeostasis [34]. Additionally, the relative abundance of two bacterial families, including *Erysipelotrichaceae* as well as *Butyricicoccaceae*, were evidenced to be enriched in the normal group. Past studies have shown that the relative abundance of *Butyricicoccaceae* was negatively correlated with diabetes [35]. Our results also suggested that the relative abundances of *Blautia* and *Butyricicoccus* in AD patients were greater than those in the normal group. Fang et al. observed that *Blautia* was significantly enriched before the administration of probiotics in AD patients, which supports our results [36]. Moreover, the abundance of *Butyricicoccus* was found to be elevated in the intestines of infants with food allergies and was reported to be positively associated with allergy rhinitis [37,38]. In addition, we found that the relative abundance of the ranked first genus *Bacteroides* was boosted significantly in Mild_AD patients. However, previous studies have reported a higher frequency of *Bacteroides* in eczema infants or a lower frequency of *Bacteroides* in eczema infants [7,39]. We speculate the inconsistent consequences may be caused by the different severities of AD patients.

Our results illustrated that the proteinogenic amino acid biosynthesis of the gut microbial community was limited, while pathways of purine nucleotide degradation metabolism were vigorous in AD patients. Interestingly, we found that the pathways, pyridoxal 5′-phosphate biosynthesis I (PRYIDOXSYN-PWY) and superpathway of pyridoxal 5′-phosphate biosynthesis, and salvage (PWY0-845) of vitamin B6 biosynthesis were severely deficient in people with severe AD. Previous studies have indicated that the microbiome in the intestine lumen converts pyridoxal phosphate into free vitamin B6, which enters the body fluid circulation through passive transport [40]. Vitamin B6 deficiency is positively related to allergy. Lack of vitamin B6 can disrupt the body’s Th1-Th2 balance, leading to excessive Th2 reactions, contributing to allergies [41].

As far as we know, this study is the first to focus on the structural and functional alterations of adult AD patients’ gut microbiomes. The limitations of this study are clear. We have to point out that the clinical information such as allergic symptoms and gastrointestinal symptoms of the included samples were declared by the participants themselves, rather than strict medical diagnosis, which might cause the results to be inconsistent compared to previous studies. In addition, we performed amplicon sequencing on fecal samples, which was only accurate to the genus level and was difficult to reveal the gut flora composition at the species level and bacterial genome. Moreover, metabolites related to AD in our results were just predictions calculated by certain algorithms. Despite this, we found some evidence to support our results, thus verification of our predictions was required. Future studies with the strict medical diagnosis of allergies and gastrointestinal symptoms, and more clinical information of participants, such as dietary habits and medication status, are warranted. Furthermore, whole genome shotgun sequencing and metabolomics sequencing of the gut microbiome should be carried out in the future, which can clarify the interaction between the gut microbiome and host in AD patients to reveal potential targets and provide novel therapeutic strategies for AD.

In summary, the composition, structure, and functional metabolic pathways of the gut bacterial community of AD patients were distinct from those of healthy people, but severe and mild AD patients had a similar overall structure of gut microbiome. We also discussed the possible interaction mechanism between the intestinal flora, functional metabolic pathways, and the host in AD patients. Gut microbiome dysbiosis and metabolic abnormalities may have essential implications for the pathogenesis of the gut–skin axis in AD patients. Further studies of metagenomics and metabolomics are warranted in order to investigate the relationship between the gut microbiome and AD and further develop beneficial treatment methods for AD.

## 4. Materials and Methods

### 4.1. Study Design and Participants

Subjects were recruited from a community trial through the collaboration between the Chinese University of Hong Kong and BioMed Technology Holdings Limited. A total of 259 participants’ fecal samples were collected, and we excluded 25 of them who were less than the age of 18. Finally, a total of 234 people, distributed between the ages of 18–68, were included in this study, among which 130 participants with healthy skin were assigned to the normal group. After atopic dermatitis diagnosis and severity assessment using the scale of EASI by the dermatologist, there were 104 patients with atopic dermatitis (AD), including 53 mild AD patients and 51 severe AD patients. Additionally, 147 of 243 participants, including 78 AD patients and 60 healthy subjects, provided the clinical information of allergy history, prevalence of diarrhea and constipation, and BMI (body mass index) for subsequent patient characteristics analyses (Appendix A). Written informed consents were obtained from all subjects or their legal guardians.

EASI, the Eczema Area and Severity Index, is a standardized scale developed in 1998 for evaluating the severity of atopic dermatitis. In this study, the dermatologist completed the EASI assessment independently [42,43,44]. Then, the mild AD group and severe AD group were classified based on the results of the EASI scale. First, the median of the EASI score of the participants was calculated. Patients whose scores were higher than the median score belonged to the severe AD group, and patients whose scores were lower than the median score belonged to the mild AD group.

### 4.2. Sample Collection, DNA Extraction, and 16S rRNA Gene Sequencing

Stool samples were homogenized in PurSafe^®^ DNA and RNA preservative (Puritan, Guilford, ME, USA) and subjected to beating with glass beads (425−600 μm, Sigma, St. Louis, MO, USA) for 1 h by following the instructions provided. Microbial DNA was isolated from the stool samples with the DNeasy Blood & Tissue Kit (Qiagen, Hilden, Germany) according to the manufacturer’s instructions. The extracted DNA concentration of each sample was quantified using a Qubit™ dsDNA HS Assay Kit (Life technology, Carlsbad, CA, USA) with a Qubit 3 Fluorometer (Thermo Scientific, Waltham, MA, USA). The amplicon library was constructed using 515F(5′-GTGCCAGCMGCCGCGG-3′)/907R(5′-CCGTCAATTTCMTTTRAGTTT-3′) primer pair spanning targeting at the V4-V5 hypervariable of 16S rRNA genes, together with adapter sequences, multiplex identifier tags, and library keys. 16S rRNA gene sequencing was performed using the Illumina MiSeq platform (Illumina, Inc., San Diego, CA, USA) following the original Earth Microbiome Project Protocols [45]. Finally, we obtained index barcodes, and adapters removed pair-end clean reads for the downstream analysis [46].

### 4.3. Microbiome Data Analysis

Microbiome bioinformatics analyses were performed using QIIME 2-2020.11, a plugin-based system, which integrates various microbiome analysis methods [47]. Briefly, quality control and a denoising filter of sequence data were processed by DADA2 [48] using the q2-dada2 plugin to obtain all observed amplicon sequence variants (ASVs) [48]. All ASVs were then aligned with mafft [49] and used to generate a phylogenetic tree with fastree2 (Price et al., 2010) for the downstream analyses via the q2-phylogeny plugin. Taxonomic assignment was performed using the q2-feature-classifier [50] plugin and a pre-trained Naive Bayes classifier, which was based on the SILVA v138 taxonomic reference database [51,52,53].

### 4.4. Functional Profiling Prediction with PICRUSt2

PICRUSt2 was developed to predict the metagenomic functional profiling of amplicon sequencing data [54]. We performed PIRCRUSt2 functions using the plugin of q2-picrust2 wrapped in QIIME 2-2019.7. LEfSe was carried out to identify the significantly enriched metagenome metabolic pathways across groups [55].

### 4.5. Statistical Analysis

Statistical analysis was conducted in R 4.0.4. Diversity analyses were performed using the R package microeco (v0.3.2) [56]. Alpha diversity was represented by the Shannon diversity index, Simpson diversity index, observed OTUs, Chao 1 richness index, abundance-based coverage estimator (ACE) index, and faith’s phylogenetic diversity index. A Kruskal–Wallis rank-sum test was performed to compare differences in alpha diversity across groups [23]. Multivariate linear regression was carried out to adjust the clinical variables and batch effects. Beta diversity was calculated based on the Jaccard distance metric, Bray–Curtis distance metric, weighted-UniFrac, and unweighted-UniFrac distance metrics. In addition, mbDenoise [28] and mbImpute [29] were performed to avoid the substantial noise from the nuisance factor such as unequal sequencing depth, overdispersion, and data redundancy. The PERMANOVA test on beta diversity (999 permutations) was applied to compare the microbial community dissimilarity across groups. The adonis function in the vegan R package was used to adjust the clinical variables and batch effects. ANCOM, analysis of the composition of the microbiome, was performed to conduct differential abundance tests across groups [57]. Linear discriminant analysis effect size (LEfSe) analysis was conducted to detect biomarkers of the gut microbiome of each group with α equal to 0.05 and an LDA score threshold of 2.0 using the online Galaxy platform (https://huttenhower.sph.harvard.edu/galaxy/) [55]. Spearman correlation was carried out to estimate the correlation between the EASI score and the relative abundance of different phylogenetics levels of gut microbial compositions. Shapiro–Wilk normality tests were carried out for normality of all data. Demographic characteristics across groups were compared using the Wilcoxon rank-sum tests (two groups) or ANOVA (three groups) for continuous variables and Chi-square tests or the Fisher exact test for categorical variables. *p*-value < 0.05 represents statistical significance.

## Figures and Tables

**Figure 1 ijms-24-12856-f001:**
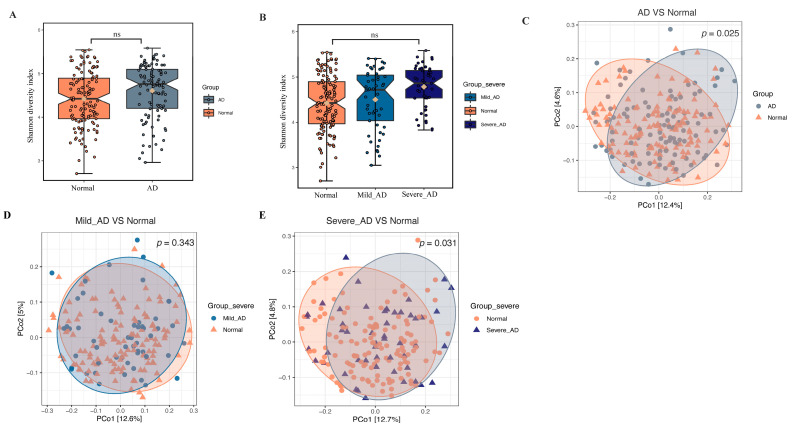
Alpha diversity and beta diversity analyses of the gut microbiome between patients with AD and healthy subjects. (**A**,**B**) illustrates the comparison of Shannon diversity index across groups. Abbreviations: ns, no significant difference was calculated by the Kruskal–Wallis test. (**C**–**E**) Beta diversity analysis across groups. Principal coordinate analysis (PCoA) plot based on unweighted-UniFrac distances, colored by groups—(**C**) AD vs. normal. (**D**) Mild_AD vs. normal. (**E**) Severe_AD vs. normal. *p* value was calculated by PERMANOVA test with permutation = 999.

**Figure 2 ijms-24-12856-f002:**
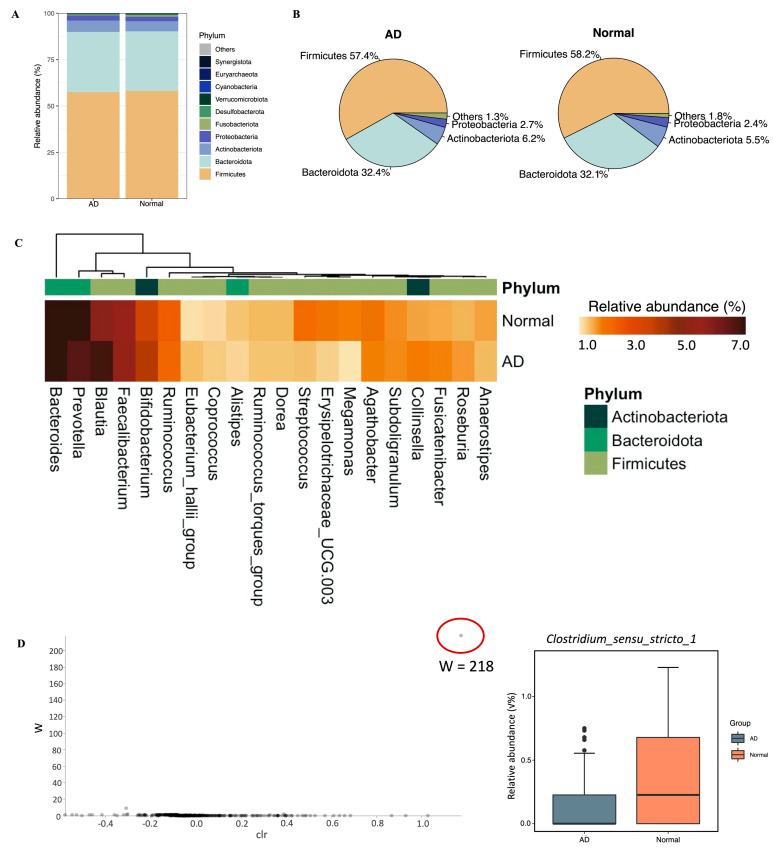
Taxonomic profiling of gut bacterial composition and differential abundant test results using ANCOM between AD and normal groups. (**A**) Bar plot and (**B**) pie charts of bacterial composition at the phylum level across groups. (**C**) Top 20 genera in the gut microbiome of AD and normal groups. (**D**) Differential abundant genus using ANCOM and boxplot of the relative abundance of *Clostridium_sensu_stricto_1*. ANCOM is used to find the differential abundant genus across groups. The *x*-axis is the clr, centered log-ration transformation on relative abundance of each SV, which means the difference in abundance of a given genus between the normal and AD groups. The *y*-axis is the W value, which is the number of times that the null hypothesis (the average abundance of a given species in a group is equal to that in the other group) was rejected for a given genus.

**Figure 3 ijms-24-12856-f003:**
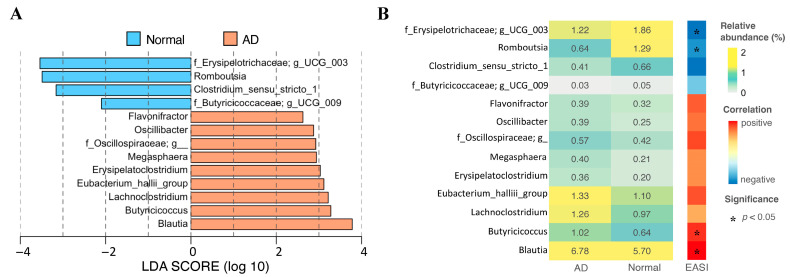
Gut microbial signatures selected by LEfSe analysis and relative abundance of gut microbial biomarkers across groups. (**A**) Bar plot of LEfSe analysis result. (**B**) Heatmap displaying the relative abundance of signatures identified using LEfSe analysis in AD and normal groups and correlation between the EASI score and the relative abundance of signatures.

**Figure 4 ijms-24-12856-f004:**
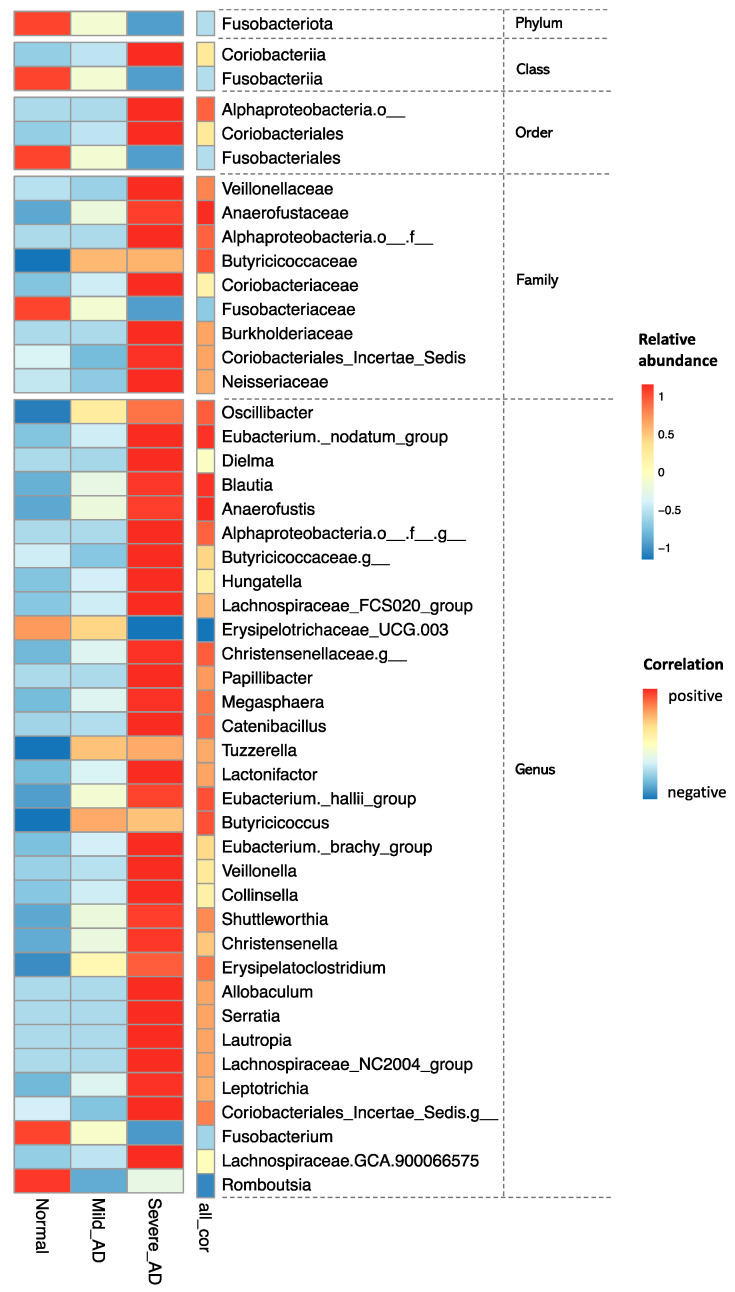
Gut microbial signatures related to AD severity at different phylogenetic levels across different groups. Each row in the heatmap is a species, each of the first three columns on the left represents a group, and the color in each grid on the left represents the average abundance of each species in each group. The fourth column indicates the significant correlations between the EASI score and the gut microbial composition at different phylogenetic levels.

**Figure 5 ijms-24-12856-f005:**
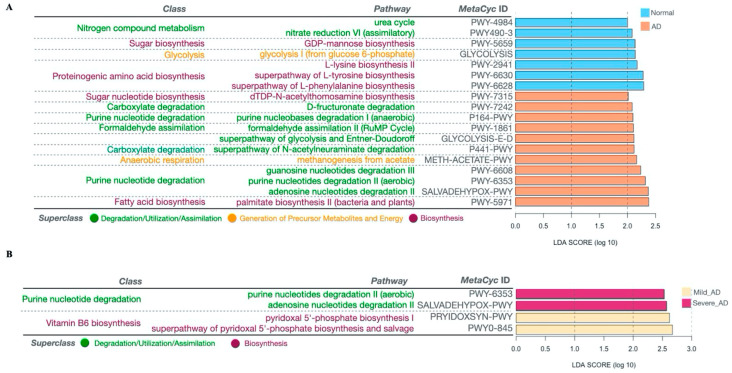
Differentially enriched predicted functional pathways across groups using PICRUSt2 and LEfSe analysis. The bar chart in (**A**) shows the LDA score (log10) of the enriched pathways between the normal and AD groups. The bar chart in (**B**) displays the LDA scores (log10) of the enriched pathways in the Mild_AD and Severe_AD groups. Two levels of annotations of each pathway are labeled, and the color of the pathway indicates the superclass to which it belongs. Green represents the superclass of degradation/utilization/assimilation; purple represents the superclass of biosynthesis; orange represents the superclass of generation of precursor metabolites and energy.

**Table 1 ijms-24-12856-t001:** Characteristics in AD group and normal group.

Parameters		AD (n = 104)		Normal (n = 130)		*p* Value ^1^
		mean±sd	n	mean±sd	n	
Age		44.3±15.1		47.7±15.1		0.216
Sex ^2^	Female		50		41	0.839
	Male		29		27	
Allergy ^2^	Yes		31		10	0.0002
	No		32		50	
	Not sure		16		8	
Food allergy ^2^	Yes		12		5	0.128
	No		65		63	
	Not sure		2		0	
Non-food allergy ^2^	Yes		17		5	0.030
	No		62		63	
Diarrhea ^2^	Yes		17		18	0.611
	No		62		50	
Constipation ^2^	Yes		26		30	0.221
	No		53		38	
BMI ^2^		23.3±3.7		23.1±3.8		0.709
Overweight (BMI > 25) ^2^	Yes		21		19	0.983
	No		52		44	
	Not sure		6		5	

^1^ *p* value was calculated using a Chi-square test and a Wilcoxon rank-sum test. ^2^ 147 of 234 participants provided the clinical information of allergy history, prevalence of diarrhea and constipation, and BMI (both height and weight) for subsequent patient characteristics analysis.

## Data Availability

The raw sequence data were deposited in the NCBI Sequence Read Archive with the accession number PRJNA778863.

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
