# Peer review of "Unique Gut Microbiome Signatures among Adult Patients with Moderate to Severe Atopic Dermatitis in Southern Chinese"

_ijms, 2023, doi:10.3390/ijms241612856_

Round 1

Reviewer 1 Report

This article deals with the analysis of a Chinese adult cohort with atopic dermatitis, this article has a fair level and presents a novelty in the field. However needs to be improved before its publication at ijms journal.

My detailed comments and suggestions are in the attached file

Authors have several typos please revise all the manuscript

Author Response

We gratefully appreciate for your careful reading, valuable comments, and constructive suggestions. We have carefully considered all comments and revised our manuscript accordingly. The manuscript has also been double-checked, and the typos and grammar errors we found have been corrected. Please kindly find the revised manuscript in the attached files. We believe that our responses have well addressed all concerns from the reviewers. We hope our revised manuscript can be accepted for publication.

Here below we summarize the major revision of our manuscript,

  1. Revise all the typos and format issues.
  2. Add ROC curves of the LEfSe results.
  3. Add the correlation analysis between the EASI score and phylogenetics level of compositions.
  4. Revise the contents of the whole article.
  5. The meta-information is available through the link provided in the article.

Reviewer 2 Report

Authors presents their observation after performing amplicon sequencing on patients with AD.

Their findings including severe deficiency of vitamin B6 biosynthesis and its positive relationship to allergy, as well as identifying genera such as Rombustsia, clostridium_sensu_strico in severe AD are important.

Did the authors ask for food sensitivities besides food allergies in their questionnaire?

In line 261, you mentioned that the we found that the child pathways of
vitamin B6 biosynthesis were severely deficient, however, study was made from adult patients please clarify.

NA

Author Response

We are grateful to your effort reviewing our paper and your positive feedback. We appreciate your accurate summary of our work and bringing forward constructive questions. We have addressed them below. We believe that our responses have well addressed all concerns from the reviewers. We hope our revised manuscript can be accepted for publication.

  1. Did the authors ask for food sensitivities besides food allergies in their questionnaire?

No, we feel sorry to say we did not include food sensitivities in the questionnaire. We didn’t realize this clinical indicator until we receive this question. This is a great idea and we decide to include “food sensitivity ” in our future related research project!

  1. In line 261, you mentioned that thewe found that the child pathways of 
    vitamin B6 biosynthesis were severely deficient, however, study was made from adult patients please clarify. 

“Child pathways” here refers to pyridoxal 5'-phosphate biosynthesis I (PRYIDOXSYN-PWY) and superpathway of pyr-idoxal 5'-phosphate biosynthesis and salvage (PWY0-845), which are the pathways belongs to pathway class vitamin B6 biosynthesis(Figure 4B). We apologize for using inaccurate word in our article which make you confused. This has been clarified in the revised version of the manuscript - “Interestingly, we found that the pathways, pyridoxal 5'-phosphate biosynthesis I (PRYIDOXSYN-PWY) and superpathway of pyr-idoxal 5'-phosphate biosynthesis and salvage (PWY0-845) of vitamin B6 biosynthesis were severely deficient in people with severe AD.” (line 276, page 11).

Reviewer 3 Report

Congratulations to the authors, it is a very interesting article, technically complete and well written, undoubtedly important in the area of dermatology. The topic is original and relevant in the field because it adds to the subject area compared with other published material new interesting findings. The references are appropriate and updated. The figures correspond to the description in the text, are well designed and reflect important information. 

I have some comments:

-Correct hyphens in the abstract (micro-biome derma-titis indi-viduals, aller-gies, etc)

-Key words are missing.

-Once the abbreviation has been defined (AD on line 34, the rest of the times that AD is referred to, should indicate AD and not atopic dermatitis (line 37, 42, 51 etc) please correct.

-Correct the design of table 1.

-Indicate whether patients at the time of sampling were taking any treatment that may influence the gut microbiome.

-Adapt the format of the references to the journal (MDPI editorial).

-The title refers to moderate-severe AD but the text does not mention moderate AD. In the manuscript reference is made to normal, mild and severe, please unify terminology.

-Is it possible to correlate the EASI scales score with the composition of the microbiome?

-Is it possible to include a ROC curve?

-Please include study limitations

Line 23. Please do not start a sentence with 'And'.

Line 146. Better to speak in the third person or impersonal, thanks.

Author Response

Thanks for your precious comments and advice. These comments are all valuable and very helpful for revising and improving our paper. We have revised the manuscript accordingly, and our point-by-point response are presented below. We believe that our responses have well addressed all concerns from the reviewers. We hope our revised manuscript can be accepted for publication.

  1. Correct hyphens in the abstract (micro-biome derma-titis indi-viduals, aller-gies, etc)

This type of hyphen misuse has been corrected.

  1. Key words are missing.

Key words are added to the article.

  1. Once the abbreviation has been defined (AD on line 34, the rest of the times that AD is referred to, should indicate AD and not atopic dermatitis (line 37, 42, 51 etc) please correct.

Abbreviation issue has been corrected.

  1. Correct the design of table 1.

The design of table 1 is corrected.

  1. Indicate whether patients at the time of sampling were taking any treatment that may influence the gut microbiome.

It is a pity to say that when we collected patient information 4 years ago, this item was not included in the questionnaire, and the original meta information form can be obtained through this link https://github.com/evy-yiweiwang999/AD_gutmicrobiome_case_control/blob/main/metainfo.csv. This is a great idea, and we will collect data of the detailed medical history and treatments received when we collect samples of volunteer participants in the future.

  1. Adapt the format of the references to the journal (MDPI editorial).

The reference format is adapted to MDPI editorial.

  1. The title refers to moderate-severe AD but the text does not mention moderate AD. In the manuscript reference is made to normal, mild and severe, please unify terminology.

The title has been modified in the revised version of the manuscript - “Unique gut microbiome signatures among adult patients with mild to severe atopic dermatitis in southern Chinese”

  1. Is it possible to correlate the EASI scales score with the composition of the microbiome?

Yes, we perform the correlation analysis of EASI score with the composition of the microbiome at different levels. Details can be found in Figure 4 and S5.

  1. Is it possible to include a ROC curve?

Thank you for your comments. We trained the classifier and plotted the ROC curves for the gut microbial signatures selected by LEfSe in FigureS4 .

  1. Please include study limitations.

The study limitations are mentioned in the second last paragraph of the Discussion part of. the manuscript(line 283-298, page 11). 

  1. Line 23. Please do not start a sentence with 'And'.

This has been modified in the revised version.

  1. Line 146. Better to speak in the third person or impersonal, thanks.

Thank you so much! This sentence was modified to express using the passive voice. 

Reviewer 4 Report

# Major comments This article by Wang et al. used 16S-rDNA sequencing to profile microbiome composition across patients with different levels of atopic dermatitis (AD) versus healthy control. The study has fairly large number of patients and control. The figure and writings are generally good. However, the depth of analyses were relatively shallow and required additional analyses and reorganization.
1. Numerous studies have shown that S. aureus is prevalent on the skin of AD patients. Additionally, non-S. aurerus Staphyloccci has been found to be related to lower risk of AD. However, no descriptions about such important fact in the introduction. Even though Staphylococci are more abundantly found in the skin, the presence of Staphylococci in the gut and the transfer between gut and skin are also evident. Thus, it would be important to show the percentage of Staphylococcus or even the S. aureus or S. epidermis if possible. The author should also point out the possible hypotheses driving studying the gut flora instead of skin flora in the introduction.   Meylan, Patrick, Caroline Lang, Sophie Mermoud, Alexandre Johannsen, Sarah Norrenberg, Daniel Hohl, Yvan Vial, et al. “Skin Colonization by Staphylococcus Aureus Precedes the Clinical Diagnosis of Atopic Dermatitis in Infancy.” The Journal of Investigative Dermatology 137, no. 12 (December 2017): 2497–2504. https://doi.org/10.1016/j.jid.2017.07.834.
Lindberg, Erika, Ingegerd Adlerberth, Bill Hesselmar, Robert Saalman, Inga-Lisa Strannegård, Nils Åberg, and Agnes E. Wold. “High Rate of Transfer of Staphylococcus Aureus from Parental Skin to Infant Gut Flora.” Journal of Clinical Microbiology 42, no. 2 (February 2004): 530–34. https://doi.org/10.1128/jcm.42.2.530-534.2004.

2. The narratives of the figures should not be organized by methodologies but by the main findings across different comparisons and correlations. For example, the Figure 1-2 should only focus on Normal versus AD, but with all different analyses in the same place. Figure 3-4, should only focus on Normal vs. Moderate AD vs. Severe AD, including PICRUST analyses. Figure 5-6 should focus on further studying how the gut microbe composition/predicted metabolic pathways associated with the EASI scores. Stratifying patients that are non-food allergy/overall allergy and study the gut microbe compositions among groups might be of great interest in this study since it is one of the major difference of between patients and healthy control.   3. A supplementary table to describe demographics of moderate AD vs. severe AD must be provided. Please make the first line of each row in the table aligned. It is so difficult to align by eyes. Further, it is important to describe that the allergy was told by patients rather than measuring IgE upfront in the table, rather than delaying such information in the discussion.   4. There are almost no descriptions of EASI score and how it was calculated, which is so important in this study. Furthermore, the score-centric analysis, based on my observation, only presents in Figure 3. There must be a dedicated analyses on this. First, this is probably closely associated with how the authors defined the moderate and severe patients. A box plot of EASI score between cohorts can be presented upfront when describing moderate AD vs. severe AD. In a new figure (e.g., Figure 5), the correlation between EASI score and different phylogenetics level of compositions can be presented. Furthermore, the correlation between predicted pathways and microbes should be presented.   5. Another immediate way to improve PICRUST analysis is to always trace back what are the driving genus/other phylogenetic levels that drive certain pathway differences. Thus, the description of predicted pathways can be linked with the differences of microbiota.
6. Please stratify patients especially focusing on non-allergy and allergy patients and perform the AD vs. Healthy control and Moderate AD vs. Severe AD. It would be of interest to see if any of these comparisons might be more standing out in stratified sub-groups.   6. It is highly recommended to supplement with metabolomics analyses to confirm some of the metabolic pathways predicted by the PICRUST.
# Minor comments - Figure 2D is very confusing. Please expands the descriptions of x-, y-axis. - Figure 2C, please shows any significant different relative abundance in the revised figureX. - In discussion, it might be relevant to point out that the CD1a-dependent T cell responses to some particular microbial PG might be important in the allergy-type responses in the patients, even though PICRUST probably has difficulties in predicting those pathways in the gut microbiome data. Further studies of lipid compositions might be of interest.   Monnot, Gwennaëlle C., Marcin Wegrecki, Tan-Yun Cheng, Yi-Ling Chen, Brigitte N. Sallee, Reka Chakravarthy, Ioanna Maria Karantza, et al. “Staphylococcal Phosphatidylglycerol Antigens Activate Human T Cells via CD1a.” Nature Immunology 24, no. 1 (January 2023): 110–22. https://doi.org/10.1038/s41590-022-01375-z.

The English is generally good but the way to lay out the narratives can be much improved!

Author Response

We feel great thanks for your professional review work on our article. As you are concerned, there are several problems that need to be addressed. According to your nice suggestions, we have made corrections to our previous manuscript, the detailed corrections are listed below.

  1. Numerous studies have shown that S. aureus is prevalent on the skin of AD patients. Additionally, non-S. aurerus Staphyloccci has been found to be related to lower risk of AD. However, no descriptions about such important fact in the introduction. Even though Staphylococci are more abundantly found in the skin, the presence of Staphylococci in the gut and the transfer between gut and skin are also evident. Thus, it would be important to show the percentage of Staphylococcus or even the S. aureus or S. epidermis if possible. The author should also point out the possible hypotheses driving studying the gut flora instead of skin flora in the introduction. Meylan, Patrick, Caroline Lang, Sophie Mermoud, Alexandre Johannsen, Sarah Norrenberg, Daniel Hohl, Yvan Vial, et al. “Skin Colonization by Staphylococcus Aureus Precedes the Clinical Diagnosis of Atopic Dermatitis in Infancy.” The Journal of Investigative Dermatology 137, no. 12 (December 2017): 2497–2504. https://doi.org/10.1016/j.jid.2017.07.834.

Lindberg, Erika, Ingegerd Adlerberth, Bill Hesselmar, Robert Saalman, Inga-Lisa Strannegård, Nils Åberg, and Agnes E. Wold. “High Rate of Transfer of Staphylococcus Aureus from Parental Skin to Infant Gut Flora.” Journal of Clinical Microbiology 42, no. 2 (February 2004): 530–34. https://doi.org/10.1128/jcm.42.2.530-534.2004.

It is a good point to look up the S. aureus or S. epidermis. Unfortunately, the amplicon sequence technique using in our paper cannot reach the resolution of species level. The only reason we didn’t mentioned S. aureus or S. epidermis is Staphylococcus is not selected as the gut microbial signature using either LEfSe or regression methods. And we also check the percentage of Staphylococcus in our results (attached file geu_EASI.csv). Among the 234 participants, Staphylococcus was detected in the intestinal flora of 11 people, except for one person whose Staphylococcus accounted for about 24%, and the Staphylococcus accounted for no more than 0.1% of the intestinal flora of the remaining 10 people.

  1. The narratives of the figures should not be organized by methodologies but by the main findings across different comparisons and correlations. For example, the Figure 1-2 should only focus on Normal versus AD, but with all different analyses in the same place. Figure 3-4, should only focus on Normal vs. Moderate AD vs. Severe AD, including PICRUST analyses. Figure 5-6 should focus on further studying how the gut microbe composition/predicted metabolic pathways associated with the EASI scores. Stratifying patients that are non-food allergy/overall allergy and study the gut microbe compositions among groups might be of great interest in this study since it is one of the major difference of between patients and healthy control.

This point has merit but the information flow of our paper was done to parallel how we wanted to present the narrative of the research.

  1. A supplementary table to describe demographics of moderate AD vs. severe AD must be provided. Please make the first line of each row in the table aligned. It is so difficult to align by eyes. Further, it is important to describe that the allergy was told by patients rather than measuring IgE upfront in the table, rather than delaying such information in the discussion.

We have added a table describing demographics of mild VS severe AD in the supplementary, see Supply table S1 for details. Additionally, you can also find the original data of the meta-information in the link https://github.com/evy-yiweiwang999/AD_gutmicrobiome_case_control/blob/main/metainfo.csv.

“Allergy history information provided by patients themselves rather than measuring IgE upfront in the table” has been added to the description of the results (line 68 page 2).

  1. There are almost no descriptions of EASI score and how it was calculated, which is so important in this study. Furthermore, the score-centric analysis, based on my observation, only presents in Figure 3. There must be a dedicated analyses on this. First, this is probably closely associated with how the authors defined the moderate and severe patients. A box plot of EASI score between cohorts can be presented upfront when describing moderate AD vs. severe AD. In a new figure (e.g., Figure 5), the correlation between EASI score and different phylogenetics level of compositions can be presented. Furthermore, the correlation between predicted pathways and microbes should be presented.

We provide more description about the EASI scale and AD severity group classification in the method section. EASI is a tool used to measure the severity of AD.

Tofte, S., et al. "Eczema area and severity index (EASI): a new tool to evaluate atopic dermatitis." Journal of the European Academy of Dermatology and Venereology 11 (1998): S197.

Hanifin JM, Thurston M, Omoto M, Cherill R, Tofte SJ, Graeber M. The eczema area and severity index (EASI): assessment of reliability in atopic dermatitis. EASI Evaluator Group. Exp Dermatol. 2001 Feb;10(1):11-8. doi: 10.1034/j.1600-0625.2001.100102.x. PMID: 11168575.

Hanifin JM, Baghoomian W, Grinich E, Leshem YA, Jacobson M, Simpson EL. The Eczema Area and Severity Index-A Practical Guide. Dermatitis. 2022 May-Jun 01;33(3):187-192. doi: 10.1097/DER.0000000000000895. PMID: 35594457; PMCID: PMC9154300.

In our study, EASI assessment was performed by a licensed dermatologist in Hong Kong. As described in the method part, we grouped patients according to the patient's EASI score and the median EASI score of the sample. Patients with a EASI score greater than the median of the overall EASI score belonged to the severe group, and vice versa belonged to the mild group.

The comparison of the EASI score across groups are showed in the attached figure. But apparently, significant differences are obtained because of the grouping method.

Additionally, we perform the analysis to select the gut microbial signatures correlated to the EASI score (Figure 4).

  1. Another immediate way to improve PICRUST analysis is to always trace back what are the driving genus/other phylogenetic levels that drive certain pathway differences. Thus, the description of predicted pathways can be linked with the differences of microbiota.

We appreciate this great idea! Unfortunately, we checked the selected pathways using PICRUST in the metacyc database. The database clearly shows the taxa known to possess this pathway, but the taxa showed in the database doesn’t appear in our gut microbial signature list.

  1. Please stratify patients especially focusing on non-allergy and allergy patients and perform the AD vs. Healthy control and Moderate AD vs. Severe AD. It would be of interest to see if any of these comparisons might be more standing out in stratified sub-groups.

That's a good point, but our allergy information is provided by patients themselves rather than a professional and clinical diagnose. Therefore, we use the adonis function in vegan R package to adjust the allergy information. Besides, if we stratify the participants based on allergy info, the sample size of each group will decrease significantly, which will bring bias to the results.

  1. It is highly recommended to supplement with metabolomics analyses to confirm some of the metabolic pathways predicted by the PICRUST.

All these samples in this study were collected four years ago (2019). And all the stool samples were used for 16S rRNA sequencing, and no samples can be used for metabolomics research. This is a great idea! We will definitely perform metabolomic analysis in our future studies.

# Minor comments - Figure 2D is very confusing. Please expands the descriptions of x-, y-axis. -Figure 2C, please shows any significant different relative abundance in the revised figureX. - In discussion, it might be relevant to point out that the CD1a-dependent T cell responses to some particular microbial PG might be important in the allergy-type responses in the patients, even though PICRUST probably has difficulties in predicting those pathways in the gut microbiome data. Further studies of lipid compositions might be of interest.   Monnot, Gwennaëlle C., Marcin Wegrecki, Tan-Yun Cheng, Yi-Ling Chen, Brigitte N. Sallee, Reka Chakravarthy, Ioanna Maria Karantza, et al. “Staphylococcal Phosphatidylglycerol Antigens Activate Human T Cells via CD1a.” Nature Immunology 24, no. 1 (January 2023): 110–22. https://doi.org/10.1038/s41590-022-01375-z.

We put more descriptions in the figure legends of Figure 2D. For more information about ANCOM analysis, please check here: https://forum.qiime2.org/t/specify-w-cutoff-for-anacom/1844/10

and also https://www.ncbi.nlm.nih.gov/pmc/articles/PMC4450248/.

Figure 2C is designed to describe the top most abundant genus. The significant different relative abundance is showed in Figure 2D when we perform the ANCOM analysis. The other gut microbial signatures are showed in Figure3.

In the discussion part, we add that the immune response to some of the particular microbial PG might play a vital role in allergy type response in the patients.

Thanks again for your comments and suggestions!

Round 2

Reviewer 1 Report

Authors need to improve their section: 

"Distinct gut microbiome composition between AD patients and healthy subjects " (lines 87-100)

Because is actually impossible to distinguish any group between samples (AD and healthy), figures 1C,D, and F do not show any clear group.

The authors are not using the Wilcoxon rank-sum test correctly.

I recommend that the authors revise the following references to improve this section.

Zeng Y, Li J, Wei C, Zhao H, Tao W. mbDenoise: microbiome data denoising using zero-inflated probabilistic principal components analysis. Genome Biology. 2022 Dec;23(1):1-29.

Busato S, Gordon M, Chaudhari M, Jensen I, Akyol T, Andersen S, Williams C. Compositionality, sparsity, spurious heterogeneity, and other data-driven challenges for machine learning algorithms within plant microbiome studies. Current Opinion in Plant Biology. 2023 Feb 1;71:102326.

Pan AY. Statistical analysis of microbiome data: the challenge of sparsity. Current Opinion in Endocrine and Metabolic Research. 2021 Aug 1;19:35-40. 

Jiang R, Li WV, Li JJ. mbImpute: an accurate and robust imputation method for microbiome data. Genome biology. 2021 Dec;22(1):1-27.

Authors need to revise any scientific name that must be in italics always

Author Response

Many thanks for your suggestions and comments. We have applied the algorithms you recommended to our data. Unfortunately, after many attempts, the distinct difference in beta diversity is still statistically significant but the scatterplot does not seem to be able to clearly distinguish the two groups. The results are in the attached figures.

We used the raw taxa-file as the input file of the mbDenoise and mblmpute. As you can see in Figures R1 and R2, it is still hard to distinguish the groups.

Although after trying different data preprocessing methods using mbDenoise and mblmpute as you recommended, it is still impossible to distinguish the two groups with the naked eye. However, no matter what data preprocessing methods we used, we can still get a statistically significant difference in beta diversity. Therefore, our conclusion of this section was not affected.

Reviewer 4 Report

The author provides reasonable justification and additional information to support their arguments. I have no further comments.

The English is generally good!

Author Response

Many thanks for your reply!

Round 3

Reviewer 1 Report

I read the new manuscript submitted by the authors, in that case, the figures R1 and R2 must be added as supporting information file.

Also, the description provided in the "authors'note" must be added in the discussion section as an opportunity area of this manuscript.

Thank you in advance for their effort and commitment with this manuscript.

Minor editing of English language required

Author Response

Thanks for your reply! 

Figures R1 and R2 have been added as supporting information file. 

The "authors' note" has been added in the discussion section.